# Signals of Potential Species Associations Offer Clues about Community Organisation of Stream Fish across Seasons

**DOI:** 10.3390/ani12131721

**Published:** 2022-07-03

**Authors:** Chen Zhang, Yuzhou Zhang, Jorge García-Girón, Kai Tan, Lei Wang, Yihao Ge, Yunzhi Yan

**Affiliations:** 1School of Ecology and Environment, Anhui Normal University, Wuhu 241002, China; zyz545520@163.com (Y.Z.); tankai0805@126.com (K.T.); leiwang@ahnu.edu.cn (L.W.); geyihao@ahnu.edu.cn (Y.G.); 2Collaborative Innovation Center of Recovery and Reconstruction of Degraded Ecosystem in Wanjiang Basin Co-Founded by Anhui Province and Ministry of Education, Anhui Normal University, Wuhu 241002, China; 3Provincial Key Laboratory of Biotic Environment and Ecological Safety in Anhui, Anhui Normal University, Wuhu 241002, China; 4Freshwater Centre, Finnish Environment Institute, P.O. Box 413, FI-90014 Oulu, Finland; jogarg@unileon.es; 5Ecology Research Unit, University of León, Campus de Vegazana, 24007 León, Spain

**Keywords:** biotic interactions, community organisation, interaction networks, latent variables, network structure, stream fish

## Abstract

**Simple Summary:**

Species interactions are one of the main factors affecting community assembly, yet the role of such interactions remains mostly unknown. Here, we investigated roles of potential species associations in fish community assembly in the Qiupu River, China. Our results suggested that potential species associations might have been underestimated in stream fish community assembly. The contribution of potential species associations to fish community assembly can be reflected by interaction network structures. Omnivorous species play an important role in maintaining network structure as they may have more associations with other species. This study highlights the importance of capturing species associations in river ecosystems across different geographical and environmental settings.

**Abstract:**

Environmental filtering, spatial factors and species interactions are fundamental ecological mechanisms for community organisation, yet the role of such interactions across different environmental and spatial settings remains mostly unknown. In this study, we investigated fish community organisation scenarios and seasonal species-to-species associations potentially reflecting biotic associations along the Qiupu River (China). Based on a latent variable approach and a tree-based method, we compared the relative contribution of the abiotic environment, spatial covariates and potential species associations for variation in the community structure, and assessed whether different assembly scenarios were modulated by concomitant changes in the interaction network structure of fish communities across seasons. We found that potential species associations might have been underestimated in community-based assessments of stream fish. Omnivore species, since they have more associations with other species, were found to be key components sustaining fish interaction networks across different stream orders. Hence, we suggest that species interactions, such as predation and competition, likely played a key role in community structure. For instance, indices accounting for network structure, such as connectance and nestedness, were strongly correlated with the unexplained residuals from our latent variable approach, thereby re-emphasising that biotic signals, potentially reflecting species interactions, may be of primary importance in determining stream fish communities across seasons. Overall, our findings indicate that interaction network structures are a powerful tool to reflect the contribution of potential species associations to community assembly. From an applied perspective, this study should encourage freshwater ecologists to empirically capture and manage biotic constraints in stream ecosystems across different geographical and environmental settings, especially in the context of the ever-increasing impacts of human-induced local extinction debts and species invasions.

## 1. Introduction

Understanding the principles governing the assembly of ecological communities is a long-standing and critical question in modern ecology [1,2]. Although new perspectives constantly converge on this issue [3,4], the emerging consensus embraces that the key processes underlying community organization are environmental filtering, species interactions (e.g., competition, predation and facilitation) and stochasticity [5,6]. These mechanisms simultaneously drive the organization of ecological communities, but their relative contributions are different across environmental gradients, spatial scales and ecosystem types [7,8].

Although many studies have attempted to disentangle the relative importance of niche-based vs. stochastic processes on community organisation [9,10], most of these studies often ignored, or even failed to empirically distinguish scenarios of species sorting by environmental vs. biotic constraints (but see García-Girón et al. [11]). However, species interactions are keystone drivers of community organisation and stability, potentially increasing the resistance and resilience of ecosystems and mediating the negative effects of disturbances [12]. Indeed, omitting species interactions can affect the inference and accuracy of conventional statistical models, thereby compromising our understanding of the mechanisms underlying community organisation [11,13].

Recently, the development of interaction network approaches (whereby species are represented as nodes and interactions as connections) has provided ecologists with a new toolbox to study how species interactions affect community dynamics [14,15]. Indeed, the topology of the resulting species interaction networks (e.g., connectance and nestedness, representing the degree of connection between species and the degree to which species-poor communities are subsets of species-rich communities, respectively) has been found to be one of the main mechanisms underlying the formation and maintenance of community structure, directly or indirectly influencing the stability of natural ecosystems [16]. For instance, habitat homogenization and food-resource limitation that are associated with increasing urbanization can reduce stream network complexity, thereby altering stream ecosystem functioning [17]. Similarly, Danet et al. [18] found an indirect influence of network topology on stream communities, with stream size and species richness modulating the connectance of fish interaction networks. This connection produced negative effects on the spatial and temporal synchrony and biomass stability of stream fish communities.

The few available empirical studies suggest that species interactions are often context dependent [19]. Indeed, interaction networks have been found to vary across space and time [20]. More specifically, variation in species abundance and functional trait composition along environmental and temporal gradients change the topology of species interaction networks, leading to high temporal and spatial heterogeneity in community structure [21]. Hence, empirical effects of species interactions on community organisation should ideally be identified with simultaneous evaluations of abiotic environmental filtering and spatio-temporal dynamics [22].

In stream ecosystems, environmental conditions (e.g., slope, discharge and flow velocity) usually co-vary from upstream to downstream areas [23]. More specifically, in subtropical streams, fish are highly adapted to the unique abiotic environment, and their communities vary significantly along longitudinal fluvial gradients [24]. Since different fish dispersal abilities across stream networks lead to diverging community organisation patterns and mechanisms [5], broad conclusions on network structure and the associated relative importance of species interactions across stream sites from different stream orders are still lacking [25]. To overcome these limitations, we investigate fish community organisation and seasonal changes in the signature of potential species associations across a mountainous stream network from the Wannan region, China. Based on latent variable models [26] and a tree-based inference method [27], we aim to: (1) partition the relative importance of abiotic environmental filtering, spatial factors and potential species associations on the assembly of fish communities across different stream orders and seasons; (2) evaluate if spatial and temporal changes in community organisation scenarios were modulated by concomitant changes in the interaction network structure.

## 2. Methods

### 2.1. Study Area

The Wannan mountainous region is located in the southern Anhui province, central China (Figure 1). The Qiupu River is one of the main water systems in the Wannan region, originating from the Xianyu Mountain and running *c.*150 km until it eventually reaches the Yangtze River. The Qiupu River Basin covers an area of more than 2200 km^2^. The average annual rainfall of the Qiupu River Basin ranges from 1400 to 1700 mm, with precipitation occurring mostly from April to September. The average annual temperature of this basin is 16 °C. As a result of the highly diverse geographical landscape and the influence of the subtropical monsoon climate, this region is considered as one of the main biodiversity hotspots in China [28].

### 2.2. Fish Sampling

In the present study, fish were sampled in the mainstream of the Qiupu River along the headstream to its mouth. Fish sampling was conducted during December 2019 (dry season) and August 2020 (wet season), respectively. The sampling sites were distributed in second-order to fourth-order streams, with a 5–8 km interval between adjacent sampling sites (Figure 1). The sites were selected based on the principle of covering different river habitats (e.g., pools, riffles) and operability. A total of 20 stream sites were sampled, including 8 sites in second-order streams, 4 sites in third-order streams and 8 sites in fourth-order streams. Due to the high complexity of the stream habitat, we used two sampling methods to ensure the most effective catch of fish [29]. A backpack electrofishing unit (CWB–2000P, China; 12 V import, 250 V export) was used to capture fish in second-order and third-order streams with water depth less than 1.5 m. The sampling was conducted using zigzag lines to cover a 50–100 m river length within 1 h at each site. On the other hand, we used drift gill nets with a mesh size ranging from 1 to 4 cm (12 m long × 1.5 m high) to capture fish in fourth-order streams, where water depth was more than 1.5 m. At each site, two drift gill nets were set, and fish were collected after the nets were drifted with the flow for 1 h. The gill nets drifted approximately 100–200 m along the stream at each site. The fish were identified to species level, counted and released back into the stream, following standard protocols (Anhui Normal University Animal Ethics Committee).

### 2.3. Local Environmental Conditions and Land Use

A suite of physicochemical and land-use covariates were measured for each stream site in December 2019 and August 2020. Specifically, eight environmental variables were measured to characterize the local habitat conditions, including elevation (m); water temperature (°C); stream width (m); water depth (m); current velocity (m/s); dissolved oxygen (mg/L); conductivity (μs/cm); and substratum coverage. Stream width was measured along three equally spaced transects across the stream. Water depth was measured at three equal interval points along each transect, and current velocity was recorded with a portable flow meter (FP111, Yellow Springs, OH, USA) at 60% of the depth of the substratum interface for each transect [30]. The water temperature, dissolved oxygen and conductivity were measured with an YSI Professional Plus meter. Based on the methods that were proposed by Kondolf [31], the stream substratum was assigned into five types, including (1) sand (<2 mm); (2) gravel (2–32 mm); (3) pebble (32–64 mm); (4) cobble (64–256 mm); and (5) boulder (>256 mm). The percentage of each substratum was evaluated with 10 transects at each sampling site.

The land-use types around each sampling site were divided into five categories (i.e., forest, grassland, urban, agriculture and waterbodies) based on a 30-m digital elevation map in ArcGIS 10.8 [30]. The percentage for each land-use type was calculated for each individual stream site.

### 2.4. Partitioning the Drivers of Fish Community Organisation

Based on fish abundance (the numbers of individuals of each species that were captured at each site) data, we applied a Markov chain Monte Carlo (MCMC) method to quantify the relative importance of the spatial factors, local abiotic environment conditions and unexplained residuals (i.e., latent variables) of fish communities for each stream order type and across different seasons. Latent variables were considered as unobserved predictors or covariates and were used to infer the biotic signals of potential species associations [22]. Note, however, that latent variables do not strictly provide evidence for proven species interactions, but that species interactions might have strong imprints on these signals [32]. Based on the ‘boral’ R package, we integrated the local environment, space, and latent variables together to build a correlated response model [26]. With explanatory (environmental and spatial covariates) and latent variables, the ‘boral’ routine fitted independent column Generalized Linear Models (GLMs) to account for any residual correlations between species. To quantify the effects of the spatial factors, we assumed that fish dispersal across each stream order was random.

Before running the models, the fish abundance data and environmental factors were standardized with Hellinger and log(x + 1) transformations, respectively [33]. In the Bayesian models, each MCMC chain was run with 40,000 iterations, a thinning rate of 30, and normally distributed priors with a mean of zero and a variance of 10 [26]. Then, we checked for convergence of the MCMC chain based on Geweke diagnostics and associated trace plots. According to the Geweke diagnostics, the Z–score exceeded 1.96 and the p-value was less than 0.05, indicating the MCMC chain did not converge. For multiple comparisons between stream orders, the *p*-values were adjusted using Holm’s method [34]. We used the predicted variance of the local environment, spatial processes and unexplained residuals from the MCMC model to infer the relative role of the environmental, spatial and potential biotic associations on fish community organization [35]. All analyses were performed in R 4.1.3 [36]. The Bayesian models were built using the function ‘boral’, and the predicted variance of each explanatory variable was calculated using the function ‘calc.varpart’, both from the ‘boral’ R package [26].

### 2.5. Constructing Poisson Log-Normal (PLN) Models

Based on environmental factors and fish abundance data, we first constructed Poisson log-normal models (a joint species distribution model) to infer the species interaction network of fish communities in different stream orders and across different seasons [37]. In addition to considering the abiotic effects, the PLN models can also avoid the detection of spurious interactions between species [27]. To control for differences in fish abundances between different sites and seasons, fish sampling efforts (i.e., the total number of fish caught at each site) were also incorporated in the models [25,38]. We combined different explanatory variables to build the PLN model (including elevation, water temperature, stream width, stream depth, current velocity, dissolved oxygen, conductivity, substratum and land-use types) and spatial factors (using geographical coordinates from which we calculated the shortest watercourse distances, i.e., the shortest distance from one stream site to another along stream corridors; *sensu* Kärnä et al. [39]. Overall, a total of 13 PLN models were built for each stream order and each season.

We evaluated the 13 PLN models and selected the best model by calculating the Bayesian information criterion (BIC) and cumulative root mean squared error (RMSE). The BIC scores explain the variational log-likelihood and number of parameters of the PLN model [27], whereas RMSE is an index indicating the predictive performance. Higher BIC values and lower RMSE scores indicate models with better fits, respectively. We selected the model with the highest BIC score and lowest RMSE scores as the best model. If more than one model showed the same BIC and RMSE values, we chose the one with the highest R^2^ value as the best model. The PLN models were run with the function ‘PLN’ of the ‘PLNmodels’ package [37,38].

### 2.6. Inferring Species Interaction Networks

We inferred fish interaction networks using a novel tree-based method (EMtree) that was proposed by Momal et al. [27], which provided network visualization based on the results of the selected PLN models. In brief, with the PLN models constructed from species abundance as a backbone, the EMtree method uses the average values of the spanning trees in the expectation-maximization algorithm to infer the undirected species interaction networks [27]. The EMtree method can infer the potential direct and indirect associations between species. In graphical models, links in the spanning trees indicated possible interactions between species, and the links were single undirected connections between the nodes. The number of interactions between all the nodes (here, species) were minimized under the EMtree graphical routine. While building the final networks, a threshold was necessary to ensure the reliability of species-to-species associations. Although a higher threshold represented higher reliability, that threshold value needed to ensure that at least one node had at least one connection [40]. Here, we selected the highest threshold between 0 and 1 to construct a network that remained connected. We first computed the stability of the frequencies for any threshold and then selected the desired stability using the function “StATS” of an EMtree package. Then, we determined the highest threshold over a desired stability value and before a node lost all connections. In addition, we iteratively resampled the network 100 times to improve its robustness.

We calculated a series of indices to describe the structure of our fish interaction networks, including connectance, nestedness, linkage density (i.e., the average number of links per species) and interaction evenness (i.e., the uniformity of links along different network pathways), all of which represented different topology aspects. To verify whether network topology contributed to the relative importance of potential biotic signals on fish community structure, we tested for correlations between network topology indices from PLN models and latent variables (i.e., unexplained residuals underlying imprints of potential species associations) from the Boral approach (see above) using the Spearman’s rank-order coefficient. The network structure indices were calculated with the ‘networklevel’ function of the ‘bipartite’ R package [41].

## 3. Results

A total of 1629 individuals were captured during both survey campaigns, representing 53 species in 12 different families (Appendix A). Cyprinidae and Bagridae accounted for 64.1% and 7.5% of the total number of species, respectively. In the dry season, 14, 21 and 26 fish species were captured in second, third and fourth order, respectively; in the wet season, 18, 25 and 31 fish species were captured in second, third and fourth order, respectively (Appendix A). *Rhynchocypris oxycephalus* and *Sinobdella sinensis* were only captured in second-order streams, whereas some piscivore species, such as *Culter alburnus*, *Culter mongolicus*, *Silurus asotus* and *Cultrichthys erythropterus*, were only captured in fourth-order stream sites. Average values and ranges of the environmental covariates for the three stream orders of the Qiupu River were shown in Appendix A.

The relative contributions of the abiotic environment, spatial factors, and residual variance differed between fish communities inhabiting different stream orders. Specifically, we found that the mean environmental and spatial factors explained 7% and 10.7% of the fish community variance, respectively. Spatial effects were pronouncedly higher in third-order stream types (17% of explained variance) than in second-order and fourth-order stream types (2% and 1% of explained variance, respectively), whereas abiotic environmental drivers accounted for more than 10% of variation in the structure of fish communities across all stream orders (Figure 2). Perhaps more importantly, we found that over 70% of variation in fish community structure could not be attributed to the observed environmental and geographical template. After Geweke’s diagnostics, Holm-adjusted *p*-values of the Z-scores for all environmental, spatial and latent variables were non-significant (*p* > 0.05), suggesting that the MCMC models converged successfully.

For fish communities of each individual stream order and season, the best fitted PLN model was selected to infer the species interaction networks (Appendix A). The PLN models with ‘site’ and ‘site’ + ‘elevation’ were chosen for the dry season, whereas PLN models with ‘site’ + ‘dissolved oxygen’ and ‘site’ + ‘flow velocity’ were selected for the wet season (Table 1 and Appendix A). We found that omnivores, such as *Zacco platypus* and *Pelteobagrus fulvidraco*, showed the highest betweenness centrality (i.e., the frequency with which one species interacts with its counterparts) for each interaction network (bigger nodes in the figures), except for those from fourth-order streams during the wet season. On the other hand, invertivore species showed the highest betweenness centrality values in second-order and third-order streams, while herbivore fish achieved the highest centrality values in fourth-order streams independently of seasonal oscillations (Figure 3).

Spearman’s rank-order coefficients showed that different network topology indices had different correlations with the latent residuals from the PLN models (Figure 4). More specifically, the connectance (*p* = 0.041) and nestedness (*p* = 0.029) were positively and negatively correlated with unexplained residuals, respectively, whereas the linkage density and interaction evenness showed no significant associations (*p* > 0.05).

## 4. Discussion

The role of potential species associations on community organisation has only recently received increasing attention [11,42]. However, due to the difficulty in directly observing species interactions in real-world ecosystems and the complexity of network inferences from abundance data [43], limited evidence still exists on how these biotic signals affect ecological communities and how they concomitantly vary across different interaction network structures. In the present study, we studied the imprints of potential species associations on fish community organisation across seasons and different stream orders. In brief, we found that these biotic couplings contributed more than environmental and spatial factors to fish community organisation. We found that community structure was also modulated by concomitant changes in the interaction networks.

In stream ecosystems, the ‘network position hypothesis’ (NPH; Schmera et al. [44]) has been confirmed by many studies (see Henriques–Silva et al. [45]). The NPH predicts that (1) environmental filtering would be the dominant mechanism underlying fish community structure in headstreams, as a result of the combined effects of isolation and environmental heterogeneity; (2) both environmental filtering and dispersal would drive community organisation in downstream areas. However, most available studies to date have ignored the imprints of potential species associations on the assembly of stream fish biotas. In addition, abiotic environmental and spatial factors have been considered as the primary processes explaining variation in stream fish community structure, despite the fact that these covariates usually only contribute a small amount of variance in traditional variation partitioning analysis [30,46,47].

Based on an MCMC modelling framework, we decomposed the effects of the abiotic environment, spatial processes and potential species associations on fish community organisation across seasons and in sites of different stream orders along the Qiupu River (China). We found that signals of potential species associations (>70% of residual variance) outweighed both the abiotic environmental component and spatial processes in structuring stream fish communities. In this sense, Mehner et al. [35] originally proposed that the unexplained residuals in Bayesian Markov chain Monte Carlo algorithms potentially indicate the imprints of species associations; they also found that species interactions have strong effects on fish community structure across European lakes. Although residual correlations are not equivalent to species interactions that are detected under manipulative in-stream experiments *per se* [32], these species-to-species associations can still provide ecological evidence of biotic constraints and their role in community organisation [35,48]. Despite this, we re-emphasize that caution is still needed when interpreting these biologically driven signals [49,50]. Our results broadly agree with Darwin’s [51] original idea that biotic interactions contribute to species distributional patterns at subtropical latitudes. For comparison purposes, the influence of these associations on stream fish communities should be further assessed and replicated in experiments and empirical studies at the community level, and across different environmental and biogeographical contexts [35]. An additional reason to anticipate further applications of this approach elsewhere is the relatively weak performance of constrained ordinations explaining compositional variation [52], particularly across freshwater ecosystems where more traditional analytical approaches account for barely 50% of the total variation [53]. These exercises are be important to prevent and foresee the consequences of anthropogenic impacts on subtropical inland waters, especially when integrating biologically driven factors in the management of local extinctions [54] and species invasions [55] along gradients of environmental degradation. Following our results, for example, the new interactions and conditions that are experienced after the addition or removal of key species are likely to influence community structure, or even contribute to the successful establishment of invaders in recipient ecosystems, facilitating some species, to the detriment of others. Hence, identifying and protecting these potential species associations is key to move forward in fish conservation, especially in the context of the highly threatened subtropical inland waters of China [56].

Various network indices have been proposed to quantify the general structure of interaction networks, depicting information on predation, mutualism and competition [57]. Network connectance, which represents the proportion of realized interactions from all possible links between the constituent species of a network, is considered as a measure of community complexity, and is predicted to modulate community stability and resilience [58]. For instance, high network connectance can increase the robustness of a network structure and protect species from secondary extinctions [59,60]. In this study, network connectance was significantly and positively correlated with the unexplained residuals underlying potential species associations, suggesting that higher connectance between species should strengthen the importance of biotic signals on community organisation. This finding might have implications beyond theoretical assumptions on the structuring mechanisms of fish communities, and suggests that a complex suite of potential species associations could confer ecosystem resiliency in streams (see Dell et al. [61] for a similar reasoning in longleaf pine ecosystems).

On the other hand, higher nestedness values suggested lower signatures of potential species associations for community organisation (Figure 4). Specifically, network nestedness provides information of a special case of asymmetrical interactions across unique components of the network [57]. A nested network structure is considered to minimize species competition in a community, thereby potentially promoting species diversity [62]. Considering correlations between network nestedness and the unexplained residuals from the Boral model, we speculate that species competition may also play a key role in structuring fish communities along the Qiupu River. This is a reasonable suggestion, not least because fish usually occupy relatively similar niches in streams, which may lead to strong competition between co–occurring species, especially during the dry season, when only small habitat patches and few food resources are available for these animals [63,64].

The distribution and frequency of different feeding habits are a proxy of potential species associations in ecological communities [11,21,65]. The species interaction network of our fish communities showed a clear trophic pattern from second-order streams to fourth-order streams along the Qiupu River. Specifically, invertivores, omnivores, herbivores and piscivores achieved their highest betweenness network centrality across different stream orders. In stream ecosystems, primary productivity in and around headwater areas is much lower than that of downstream sites. Hence, the trophic structure of stream fish usually shows a longitudinal invertivore–omnivore–herbivore–piscivore replacement pattern from upstream to downstream areas [66], a pattern that was mostly confirmed by our Bayesian modelling approach. Interestingly, however, we found that omnivorous fish showed the highest betweenness network centrality across seasons and in sites from different stream orders. Network centrality indicates the importance of species for the interaction network structure and is a common measure when it comes to identifying ‘keystone species’ in ecosystems [67]. In our stream sites, many omnivorous species may have competed for food with the remaining fish feeding groups, which probably led them to play a key role in maintaining the stability of most interaction networks [68].

## 5. Conclusions

Based on latent models and a tree-based method, we studied the relative contribution of abiotic environment filtering, spatial factors and imprints of potential species associations on fish communities across seasons and in different stream orders. We found that biotic signals reflecting potential species associations might have been underestimated in classical community-based assessments of stream fish. For instance, the high betweenness network centrality of omnivorous fish suggests that competition between different trophic guilds may have a primary role in maintaining the seasonal stability of fish interaction networks in streams. From an applied perspective, our findings emphasize that identifying and protecting these potential species associations networks is key for the conservation and management of stream communities, especially in the context of the ever-increasing, human-induced invasions and local extinctions of key species in Subtropical Asia. We encourage future studies to capture biotic constraints as potentially important long-term assembly mechanisms in running waters, and to exceed more traditional analytical approaches that only consider species sorting by abiotic environmental conditions.

## Figures and Tables

**Figure 1 animals-12-01721-f001:**
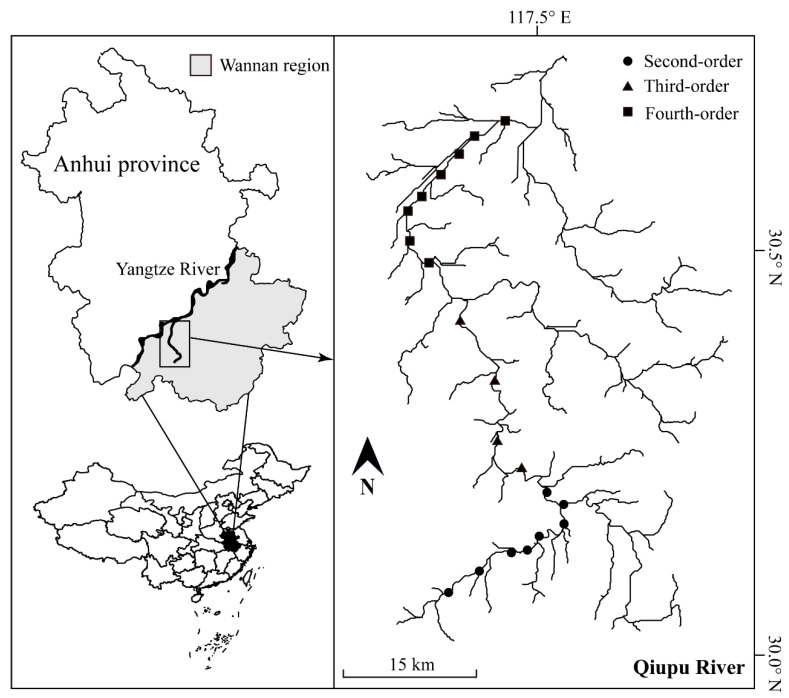
Map of the fish-sampling sites across the Qiupu River in Wannan region, which is located in the southern Anhui province.

**Figure 2 animals-12-01721-f002:**
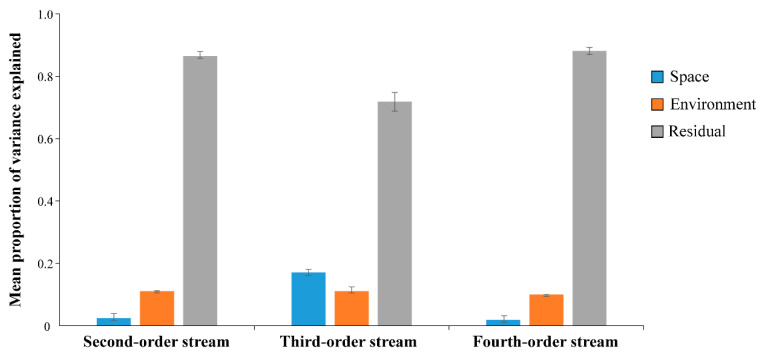
Relative contributions (i.e., mean values averages across species seasons) of abiotic environmental conditions, spatial factors and unexplained residuals underlying potential species associations to variation in fish community composition for different stream orders.

**Figure 3 animals-12-01721-f003:**
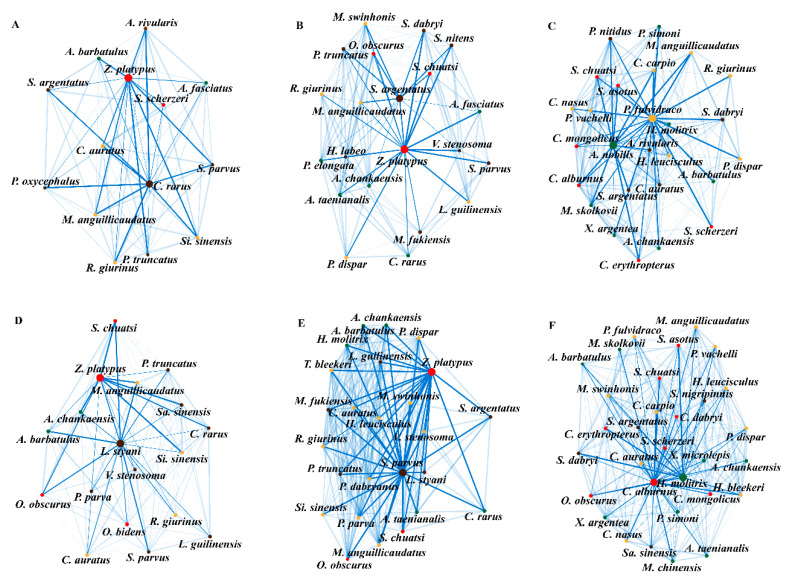
Fish interaction networks of the Qiupu River for second-order to fourth-order stream sites during the dry (**A**–**C**) and wet (**D**–**F**) seasons. Each edge has a width proportional to its conditional probability. Node colours correspond to species showing different feeding habits: (1) yellow: omnivore; (2) red: piscivore; (3) brown: invertivore; and (4) dark green: herbivore.

**Figure 4 animals-12-01721-f004:**
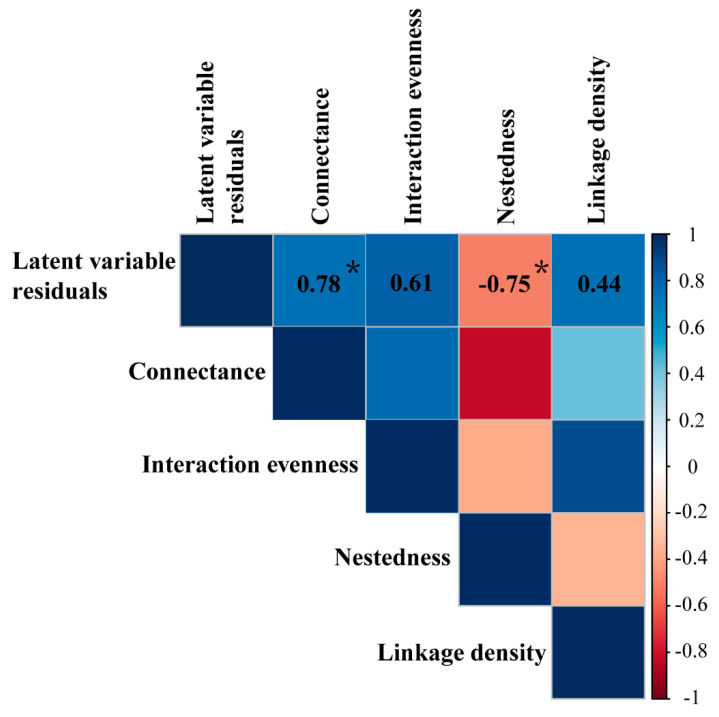
Spearman’s correlations between unexplained residuals underlying imprints of potential species associations (i.e., latent variable residuals) and topology indices of network structures. Positive and negative correlations are shown in blue and red, respectively. Only the correlations of latent variable residuals and topology indices of network structures were shown. Significant results were presented with *.

**Table 1 animals-12-01721-t001:** Model fit diagnostics and prediction errors for each Poisson log-normal (PLN) models.

Season	Stream Orders	Selected Variables	BIC	RMSE	R^2^
Dry	Second-order	Site + elevation	−267.714	0.567	0.946
	Third-order	Site + elevation	−256.428	0.235	0.945
	Fourth-order	Site	−557.069	0.338	0.770
Wet	Second-order	Site + dissolved oxygen	−385.017	0.487	0.875
	Third-order	Site + velocity	−330.697	0.032	0.932
	Fourth-order	Site + dissolved oxygen	−820.550	0.253	0.908

Abbreviations: BIC, Bayesian information criterion; RMSE, cumulative root mean squared error; velocity, stream velocity.

## Data Availability

The fish list is stored in the Figshare repository (Available online: https://doi.org/10.6084/m9.figshare.17694260.v1, accessed on 27 December 2021).

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
