# Peer review of "Signals of Potential Species Associations Offer Clues about Community Organisation of Stream Fish across Seasons"

_animals, 2022, doi:10.3390/ani12131721_

Round 1
Reviewer 1 Report
Overall, I found this to be a well-written paper describing an interesting analysis on stream fish communities in the Quipu River (China). My main question regarding this work is how much of the species interactions are related to individual species associations with habitat variables versus interactions with one another? It seems like the results could be confounded by habitat associations. Two species occupying the same habitats could be competing for the same resources, but it is unclear how the modeling approaches used could account for correlations in habitat use relative to mutualistic or competitive/predatory interactions. Alternatively, the authors may consider adjusting the language to emphasize that these are "associations" as "interactions" suggests to me that the authors have more fine-scale information than are presented in this work. Further, the authors may consider clarifying how the three analyses are related in the methods. It was unclear if the analysis described in 2.4 and 2.5 were achieving the same goals or if 2.5 was required to complete the tasks in 2.6 to visualize results. What does the analysis in 2.4 provide that 2.5 does not? Also, model equations would have been helpful to understand the model structure. I am also skeptical of the statements in the text that unexplained residuals represent biological interactions. It seems possible variation is uncaptured by the environmental factors and there could be unexplained variation from other environmental factors, sampling error, or randomness. Also confounding factors such as species richness could confound interpretation of the results. Although the analyses employed are interesting, if network connectance is correlated with species richness, the work could be simplified by examining the relationship of richness and residuals from the analysis in 2.4. How could factors such as species richness affect the results?
Line 12-18: The authors did not provide a simple summary as this text appears to be guidance from the journal.
Line 28-29: It is unclear to me what this sentence means. here. For instance, does this mean that omnivores were associated with many species across stream orders?
Line 31-34: It seems this sentence needs commas before "potentially" and and after "interactions"
Line 54: Consider replacing "Although many attempts have already tried.." with "Although many studies have attempted.."
Line 73: Replace "to" with "with" and remove "impacts"
Line 79: Consider replacing "evidence suggests" with "studies suggest"
Line 117: The phrase "Fish campaigns were done" reads somewhat awkwardly to me. Consider replacing with "Fish sampling was conducted"
Line 118-120: Please describe how sampling sites were selected.
Line 122-123: Please clarify how using two gears ensured "the most effective catch of fish". Using gill nets in higher stream order sections of the river could bias catches toward different sizes of fish across stream orders.
Lines 128-130: The authors might consider providing a summary of how much distance was covered by drifting gill nets. Gill nets drifting greater distances could catch greater numbers of fish by covering more space, which could be confounded with environmental correlates such as water velocity.
Line 152: "drives" should be "drivers"
Line 167-169: Why did the authors choose to transform the data rather than use distributions that are better suited for count data (e.g., Poisson)?
Line 169-171: Did the authors conduct a sensitivity analysis to determine if their prior specifications influenced model fit?
Line 172-174: The authors indicate that "the MCMC chain did not converged" - Would this indicate that the models are not suitable for inference?
Line 174: What was compared in the "multiple comparisons"? Please clarify. Also, please provide a reference for the Holm's method.
Line 175: Is there a reference to support the validity of using this approach to assess the relative role of these various factors? How is this approach superior to other possible approaches?
Line 179: Please provide the version and citation for R.
Line 186-189: Please clarify the importance of controlling for differences in abundance between sites and seasons. It is unclear to me how this differs from the response variable. Were site and seasons included as a level of a hierarchical model?
Line 194-195: The authors note that the suite of models was developed for each stream order and season. What barriers exist to developing a model that incorporates stream order and season? Is there information lost by not incorporating data across time and space?
Line 196: The authors should provide support for using BIC and not another (or multiple) information criterion (criteria).
Line 199: Lower BIC values indicate a better model as the calculation of BIC includes a negative log-likelihood term.
Line 202-203: If BIC and RMSE indicate two multiples are equivalent, why not use a model averaging approach to make inference? This suggests to me that the candidate models likely both have merit and should be considered.
Line 206: During my initial read of the paper, I interpreted this as an additional analysis but it seems to be a data visualization procedure using PLN results. The authors may consider clarifying the purpose of this part of the study in the topic sentence.
Line 211: The text says "see below", but is not specific about where the referenced figures or text can be found.
Line 222: Is there a value or figure you can provide to support that the results "converged" or were stable at the level of 100 iterations? I imagine this is a computationally demanding process, but I generally see considerably larger numbers of iterations for resampling procedures.
Lines 234-239: The summary of the dataset here seems to be somewhat brief. The authors may consider adding some additional information. I also was unable to find Table S1 in the reviewer website, so I am unsure how much information the table provides to supplement the text.
Line 241-247: The authors note that environmental/spatial effects had a relatively low contribution to the variance explained. The text here makes me wonder if increasing the contrast of the spatial/environmental factors by examining the communities across stream orders (i.e., incorporating different stream segments into a single analysis) would improve the analysis and the amount of variation that could be explained. This might provide an examination of factors affecting communities across spatial scales. Stream order could be a better predictor for community composition than those examined.
Line 247-250: Is this part of the results section stating that 70% of the variation is unexplained? The wording here could be confusing to readers as it seems to suggest this variation is accounted for by the models, but it appears that it is unexplained and could be related to unexamined factors or those examined but not captured in the model.
Line 257-267: Given that the methods section notes that BIC is misapplied, I am uncertain how to interpret these results as finding the model with the greatest BIC would select the worst model. Providing a table of all the models with their BIC and RMSE values would be helpful. It was also unclear to me what "betweenness centrality" means. The authors provide a definition which was helpful, but it is unclear what a "counterpart" is (i.e., is this other species in the same feeding guild, other species occupying the same habitats, etc.?). Does high betweenness centrality mean that omnivores tend to interact with only other omnivores or they interact with the most other species? Please clarify.
Line 258-260: Is this specific to a particular stream order?
Line 269-272: Figure 3 and its caption are presented on different pages. Please place figures and captions on the same pages throughout.
Line 293-294: How do we know that these relationships are species interactions and not species responding to the same environmental factors or occupying the same habitats? The authors might consider providing some additional support that the model can disentangle correlations in habitat use and species interactions.
Lines 345-356: Is there a relationship between network connectance, species richness and the residuals? This part of the manuscript made me wonder if network connectance could be driven by the number of species which seems like it could potentially confound the interpretation of the results. I suspect greater numbers of species would result in greater "strain" on model fitting and potentially greater residuals.
Figure 1: Including a larger map showing the location/boundaries of central China and the Wannan region (mentioned in the introduction) in relation to China as a whole would be beneficial for international readers. Further, the authors might consider providing a box around the section of the Qiupu River that defines the area shown in the panel on the right. The authors may also consider adding additional details on the contents in the figure in the caption, such as noting the Anhui province and sampling time frames. Further, the authors may consider adding geographic references made in the text to their map(s) (e.g., Xianyu Mountain).
Figure 2: How was uncertainty calculated for the error bar values plotted in the figure? Is uncertainty from the individual models incorporated or are these related to a standard error from the calculated means?
Figure 4: This figure seems overly complicated. It is unclear what the sizes of the circles mean. The authors could likely just provide the correlation values in the cells and denote those that are significant using an a defined symbol.
Reviewer 2 Report
This is an interesting paper studying fish species interactions in streams of different orders in China rivers.
I don’t have major problems with this paper, despite the fact that I am not a specialist in this king of statistical analysis and this should be reviewed by a statistician.
However, as an ecologist, I found that 40 fish by station is a small sample size to study the interactions between species especially considering that there is as many as 53 potentials species in the river. What is the % of potential diversity in your data compare to the diversity expected? This could be important in term of interactions and community structure in your samples.
Figure is not clear of what is significant and not. This should be clarified or the figure removed. What are the confounding effects? For example Nestedness and Interaction evenness?? This has to be clarified.
Paper can be accepted only after these clarifications.
